

# Evolutionary algorithms guided by Erdős–Rényi complex networks

Víctor A. Bucheli[1], Oswaldo Solarte Pabón[1] and Hugo Ordoñez[2]

[1] Escuela de Ingeniería de Sistemas y Computación, Universidad del Valle, Cali, Valle, Colombia
[2] Departamento de Sistemas, Universidad del Cauca, Popayán, Cauca, Colombia

## ABSTRACT

This article proposes an evolutionary algorithm integrating Erdős–Rényi complex networks to regulate population crossovers, enhancing candidate solution refinement across generations. In this context, the population is conceptualized as a set of interrelated solutions, resembling a complex network. The algorithm enhances solutions by introducing new connections between them, thereby influencing population dynamics and optimizing the problem-solving process. The study conducts experiments comparing four instances of the traditional optimization problem known as the Traveling Salesman Problem (TSP). These experiments employ the traditional evolutionary algorithm, alternative algorithms utilizing different types of complex networks, and the proposed algorithm. The findings suggest that the approach guided by an Erdős–Rényi dynamic network surpasses the performance of the other algorithms. The proposed model exhibits improved convergence rates and shorter execution times. Thus, strategies based on complex networks reveal that network characteristics provide valuable information for solving optimization problems. Therefore, complex networks can regulate the decision-making process, similar to optimizing problems. This work emphasizes that the network structure is crucial in adding value to decision-making.

## INTRODUCTION

Evolutionary computation is a branch of artificial intelligence (*Fogel, 1998*) that aims to enhance optimization algorithms, improving solution quality, convergence, and other performance measures (*Michalewicz, 1996*). This article introduces an evolutionary algorithm integrating Erdős–Rényi complex networks to guide population crossovers dynamics of the population or solutions, enhancing candidate solution refinement across generations. This article addresses the Traveling Salesman Problem (TSP), a well-known optimization problem integrating an evolutionary algorithm to an Erdős–Rényi Complex Network. The underlying hypothesis is that a strategy guided by a complex dynamic network, specifically the Erdős–Rény complex networks, will yield better performance compared to a traditional algorithm. Previous studies highlight the integration of evolutionary algorithms with complex networks (*Zelinka et al., 2010*; *Triana, Bucheli & Solarte, 2022*; *Zelinka et al., 2011*, *2010*, *2014*). Moreover, *Llanos-Mosquera et al. (2022)* have suggested that a strategy employing the selection of individuals guided by Watts–

Corresponding author
Oswaldo Solarte Pabón,
oswaldo.solartep@alumnos.upm.es

Strogatz network models can outperform traditional evolutionary algorithms in solving the TSP.

The empirical findings demonstrate that the proposed model exhibits robust convergence capabilities and significantly reduces execution times when compared to other models and the baseline traditional algorithm. The evaluation of the proposed approach involved a selection of Traveling Salesman Problem (TSP) benchmark instances, characterized by varying city quantities spanning from 5 to 1,000. In order to address algorithmic variability, a series of experiments with diverse random seeds were systematically conducted to account for stochastic effects. The reported results are derived from aggregating outcomes across more than 100 independent runs, offering a comprehensive assessment of performance. Evaluation criteria encompassed key metrics, including average fitness values and convergence speed.

Results show the performance of the proposed Erdős–Rényi network approach compared to the other algorithms. These findings consistently indicate the proposed model's effectiveness, even surpassing the Watts–Strogatz network approach. Additionally, we assessed the computational cost of the proposed model compared to the traditional strategy. The results indicate that as the size of the instance increases, the computational cost of the proposed models remains acceptable, demonstrating a comparative advantage over the other evaluated models.

The 'Materials and Methods' section provides an overview of the evolutionary computation and complex networks model, offering a comprehensive review of previous works. It also outlines the Traveling Salesman Problem (TSP) and introduces the proposed model. 'Experimental Settings' presents the experimental results, while the 'Discussion' section discusses and concludes the research and suggests potential directions for future work.

## MATERIALS AND METHODS

Since its development in 1956 (*Friedman, 1956*; *Bremermann, 1958*), evolutionary computation has become one of the most widely used techniques for solving optimization problems. According to *Holland (1992)*, evolutionary algorithms are a family of search and optimization methods inspired by biological evolution. In 1976, the evolutionary computation was understood as an evolution strategy, which is an optimization technique based on ideas from natural evolution (*Kellermayer, 1976*). Evolution strategies use natural representations that depend on the problem and primarily employ recombination, mutation, and selection as search operators for optimization processes. Over the past few decades, bio-inspired computation has garnered significant attention as a branch of artificial intelligence. It has been extensively studied to achieve high performance across a wide range of complex academic and real-world problems (*Vikhar, 2016*; *Del Ser et al., 2019*).

### Complex networks in evolutionary computing

A comprehensive review of complex networks in evolutionary computation has been conducted by *Jiang, Liu & Wang (2016)*, while *Pizzuti (2017)* provides a systematic review

of evolutionary algorithms that incorporate complex networks. These works offer an overview of how complex networks can be integrated into evolutionary computation. In the study conducted by *Zelinka et al. (2010)*, differential evolution is employed along with the SOMA algorithm to model complex networks, representing parent-child relationships as networks. Other related works (*Zelinka et al., 2010*; *Zelinka, 2011*; *Zelinka et al., 2014*) have also explored the integration of complex networks into evolutionary algorithms. These works discuss various aspects, including the analysis of degree distribution to investigate premature convergence, the average path length between nodes for identifying complex networks, the clustering coefficient for monitoring individual progress in each iteration, and centrality between nodes as an alternative method for identifying the best individuals.

Furthermore, experimental results have shown the integration of complex networks into evolutionary algorithms applied to different problems such as the Knapsack Problem (*Triana, Bucheli & Solarte, 2022*) and optimization problems like the Ackley function, Bealen function, Camel function, and Sphere function (*Triana, Bucheli & Garcia, 2020*). Several studies have focused on exploring the properties of complex networks in relation to solving optimization problems. Additionally, there have been investigations centered around community detection (*Liu, Liu & Jiang, 2014*) and PageRank distance (*Jiang, Liu & Wang, 2016*). Moreover, according to *Llanos-Mosquera et al. (2022)*, properties of small-world complex networks, such as the clustering coefficient and average path length, have been evaluated.

## The traveling salesman problem

The Traveling Salesman Problem (TSP) is a well-known combinatorial optimization problem in computer science (*Applegate et al., 2006*). It involves finding the shortest possible route that a salesman can take to visit a set of cities and return to the starting city while visiting each city exactly once. The significance of the TSP lies in its applications in various domains, including transportation, logistics, network design, and operations research, where finding efficient routes and minimizing travel costs are crucial for improving resource allocation, minimizing expenses, and optimizing overall system performance (*Laporte, 1992*).

The TSP is an NP-hard problem, meaning that there is no known algorithm that can solve all instances of the problem in polynomial time. Therefore, various heuristic and metaheuristic algorithms have been developed to approximate solutions for large-scale instances of the problem. The importance of the TSP extends beyond its theoretical complexity. It has practical applications in various domains, such as transportation planning, logistics, circuit board manufacturing, and DNA sequencing. Finding efficient solutions to the TSP has significant implications for optimizing resource allocation, minimizing costs, and improving overall operational efficiency in real-world scenarios (*Kanna, Sivakumar & Lingaraj, 2021*).

Formally, the TSP can be defined as follows: Given a complete undirected graph with n vertices representing the cities, where each edge between two vertices represents the distance or cost of traveling between the corresponding cities, the objective is to find a

Hamiltonian cycle of the minimum total cost. A Hamiltonian cycle is a cycle that visits each vertex exactly once and returns to the starting vertex (*Ortiz-Astorquiza, Contreras & Laporte, 2015*).

The Traveling Salesman Problem (TSP) can be mathematically described as follows:

Let G = (V, E) be a complete undirected graph, where V is the set of cities and E is the set of edges representing the connections between the cities. Each edge e = (*u, v*) has a non-negative weight or cost w(e) associated with it, representing the distance or travel cost between cities *u* and *v* (*Applegate et al., 2006*). We aim to find a Hamiltonian cycle C in G that visits each vertex in V exactly once and returns to the starting vertex. A Hamiltonian cycle is a closed path that passes through every vertex in a graph exactly once, except for the starting and ending vertices which coincide.

The objective function of the TSP (F) is to minimize the total cost or length of the Hamiltonian cycle C. This can be represented as the sum of the weights of the edges in C:

$$F = \text{minimize} \sum w(e), \text{ for all } e \in C \tag{1}$$

Subject to:
- *Each vertex v ∈ V must be visited exactly once.*
- *The cycle C must be closed, i.e., it starts and ends at the same vertex.*
- *The cycle C must not contain any subcycles, i.e., it must be simple.*

## A proposed model based on evolutionary model and Erdős–Rényi networks

### Model Erdős–Rényi network

According to *Erdós & Rényi (1960)* the creation of a network is a stochastic process for generating random graphs, with two parameters:

- Network size (n) determines the size of the network under consideration, it is the number of nodes (vertices) within the network, and it is denoted as 'n'.
- Specify connection probability (p), this parameter governs the overall connectivity of the resulting network, the probability 'p', which characterizes the likelihood of a connection (edge) existing between any pair of nodes in the network.

The model systematically examines all possible pairs of nodes to determine potential connections within the network. For each pair, a random number is generated from a uniform distribution between 0 and 1. A connection between the nodes is formed if this random number is less than or equal to 'p'. Conversely, no connection is established if the generated number exceeds 'p'. This iterative process results in a random network adhering to the Erdős-Rényi model.

### Proposed algorithm

According to *Osaba et al. (2018)*, the Traveling Salesman Problem (TSP) is commonly addressed using heuristic algorithms. Previous research has proposed several complex network properties in relation to the TSP, such as Community Detection (*Liu, Liu & Jiang,*

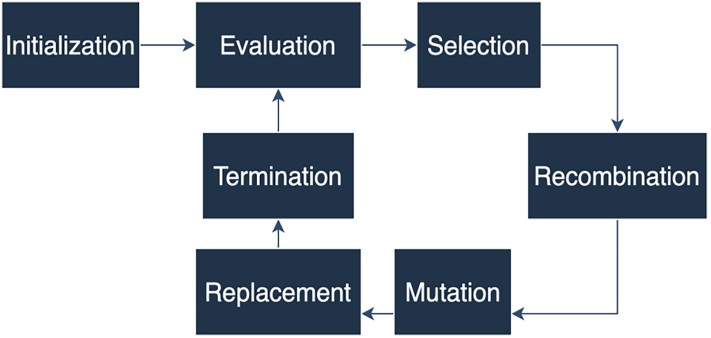

**Figure 1  A genetic algorithm diagram.**               

*2014*) and PageRank distance (*Jiang, Liu & Wang, 2016*). Furthermore, evaluations have been conducted on the properties of small-world complex networks, including the clustering coefficient and average path length (*Triana, Redondo & Bucheli, 2019*).

An evolutionary algorithm can be described as follows: Firstly, a population of solutions is created, followed by a crossover, selection, recombination, mutation, replacement, and finalization phase (*Yu & Gen, 2010*). The key components of an evolutionary algorithm include initialization, where a population of individuals is randomly generated as the initial set of candidate solutions. Evaluation is performed for each individual in the population using a fitness function that measures their quality or performance.

Selection involves choosing individuals with higher fitness values to become parents for the next generation, often based on a probabilistic mechanism like roulette wheel selection or tournament selection. The selected individuals are then combined to produce offspring through recombination or crossover operators. The last critical process is recombination, which simulates genetic recombination occurring during reproduction in biological systems. Similar to genetic processes, offspring are subjected to random changes or mutations, introducing new variations into the population. Finally, the new offspring replace some individuals in the current population, ensuring a constant population size. The algorithm terminates when a stopping criterion is met, such as reaching a maximum number of generations or finding a satisfactory solution (see Fig. 1).

The proposed model incorporates the Erdős–Rényi network into the initialization, selection, and recombination processes. In contrast, the traditional algorithm relies on randomness for these three processes. In the proposed algorithm, the initialization process entails defining all parameters, such as population size, mutation rate, and others, followed by the random creation of the population of solutions.

Figure 2 illustrates the Erdős–Rényi process, where a Model Erdős–Rényi network is simulated with a size 'n' equal to the previously defined population size and 'p' set to 0.5. Additionally, each solution is randomly assigned to a node within the Erdős–Rényi network, contributing to the novel configuration of the ensemble of solutions.

While in the traditional algorithm, these processes are driven by randomness, in the proposed algorithm, they are influenced by the network's structure. Consequently, the selection and recombination of the population are guided by the Erdős–Rényi network. In

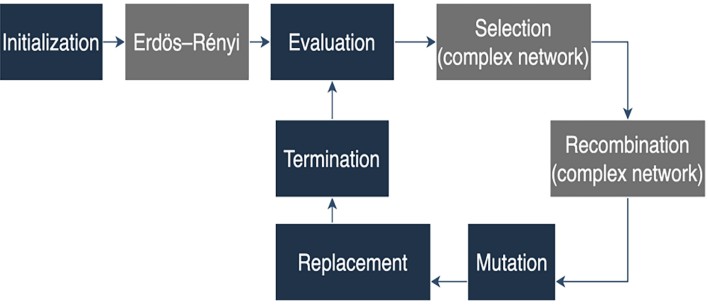

**Figure 2  Proposed algorithm diagram.**

---

**Algorithm 1** The pseudocode of proposed model for TSP

---

*Population* ← pseudo randomly generated population of size n
*Erdős–Rény networks* ← Each node associated to a unique individual of population
**while** *stopCriterionNotReached* **do**
    **for** $i \leftarrow 1$ to n **do**
        *individualA* ← i individual of population
        **for** $i \leftarrow 1$ to k **do**
            *neighborsA* ← set of all *individualA*'s neighbors
            **if** length(*neighborsA*) $> 0$ **then**:
                *individualB* ← choose one individual from *neighborsA* by a given the best criterion
                *Reproduction Stage* with *individualA* and *individualB* to generated a new
*individualC*
                *Mutation Stage* with *individualC*
                **if** (*individualC.fitness*) $>$ (*individualA.fitness*) **then**:
                    Replace *individualA* with *individualC* in population
                **end if**
            **end if**
        **end for**
    **end for**
**end while**

---

**Figure 3  The pseudocode for generation of individuals for TSP.**

---

this approach, individuals from the population are chosen as parents for the next generation based on the connections of one node in the network; it is named as (*k*) in the algorithm (see Fig. 3). The selection probability of each individual is not proportional to its fitness but rather linked to the node degrees within the Erdős–Rényi network. Additionally, the recombination applies crossover operators to the selected parents, combining the ordering of cities from two parents to generate new solutions. In summary, while the traditional algorithm relies on random selection, the proposed algorithm integrates the Erdős–Rényi network's structure to guide these processes. Figure 2 shows the proposed algorithm with the Erdős–Rényi network.

### Application of the proposed algorithm to the Traveling Salesman Problem

In the context of solving the TSP using a traditional evolutionary algorithm, each individual in the population represents a unique city visitation order. The process begins with the initialization of individuals, where random permutations of cities are created. Fitness evaluation involves calculating the inverse of the total distance traveled, with higher fitness indicating shorter paths, where the fitness values are computed by de fitness

function (F), (see Eq. (1)). Selection probabilities are based on fitness, and recombination combines city orderings from selected parents using methods like order-based crossover for TSP. Mutation introduces random changes, maintaining diversity. The current population is replaced by offspring, and the algorithm continues until a predefined stopping criterion is met, with iterative execution of evaluation, selection, recombination, mutation, and replacement steps. The traditional evolutionary algorithm approach to solving the Traveling Salesman Problem (TSP) relies heavily on randomness throughout its major processes. In contrast, alternative solutions employ network models like the Watts and Strogatz model, the Barabási–Albert model, or employ the PageRank algorithm, parallelization, or tailored modifications to enhance the TSP-solving process. These diverse approaches collectively represent the current state of the art in addressing the TSP using evolutionary algorithms.

This article concentrates on incorporating complex networks, specifically Erdős–Rényi networks (*Erdós & Rényi, 1960*), into individuals' crossover and selection process. The Erdős guide the population of solutions to the optimization problem–Rény network model. The research question investigates the performance of a traditional evolutionary algorithm that utilizes Erdős–Rényi complex networks for solving the Traveling Salesman Problem.

Moreover, the crossover process of the population is represented using complex networks, where a graph captures the parent-child relationships and the groups of solutions or clusters of solutions. The vertices (nodes) in the graph correspond to individuals or the population of solutions, while the edges (links) represent the connections between individuals. To enhance the quality of the solutions, links are established between existing nodes in the population, following the structure of the Erdős–Rény model. In this proposed approach, parents are selected according to the Erdős–Rény network. Thus, Parents are selected based on the characteristics of the neighborhood surrounding that individual in the network. Maintaining the better solutions and increasing the crosses between better solutions by the parent-child relationships and the groups of solutions or clusters of solutions. Consequently, parents are not chosen randomly but rather based on the topology of the Erdős–Rény network. By incorporating this modification into the traditional algorithm, which typically selects solutions randomly for crossover, improved convergence and shorter execution times were observed.

The pseudocode for the evolutionary algorithm integrating Erdős–Rény networks is presented in Fig. 3. The algorithm uses the best neighbor model, in which it consists in the selection of the neighbor through the roulette method. The intention is to favor the selection of the neighbor with better performance.

The TSP is acknowledged as NP-hard, yet its practical applications underscore the significance of an effective solution. As a term within computational complexity theory, TSP represents a nondeterministic polynomial-time (NP) problem, which can be addressed by nondeterministic polynomial-time bounded Turing machines. Notably, the Genetic Algorithm (GA) is widely recognized as a vital meta-heuristic technique for resolving intricate large-scale optimization problems (*Dao & Marian, 2013*; *Paul et al., 2015*; *Shapiro & Delgado-Eckert, 2012*). Despite advancements, solving extensive TSP

instances remains a daunting and unresolved challenge, characteristic of other NP-hard problems, due to its computational intractability. TSP solution strategies are broadly divided into exact methods and meta-heuristic approaches. Although meta-heuristic methods cannot ensure optimal TSP solutions, they demonstrate efficiency in producing satisfactory solutions, particularly for large-scale TSP scenarios. Consequently, meta-heuristic methods, including the extensively used GA, are often preferred for addressing such complex optimization problems.

The computational complexity of Genetic Algorithms (GAs) in an academic context depends on various factors, including the genetic operators, their implementation (which can significantly impact overall complexity), the representation of individuals, the population size, and the fitness function. In typical scenarios using point mutation, one-point crossover, and roulette wheel selection, the complexity of GAs is based on the number of generations, the population size, and the size of individuals (*Nopiah et al., 2010*). It's important to highlight that this analysis excludes the complexity associated with the fitness function, which varies based on the specific application.

In the context of solving complex optimization problems, GAs are compared to other solution methods, such as Brute Force, Dynamic Programming, and Branch and Bound. These methods have their own complexities, with Brute Force being $O(n!)$, Dynamic Programming $O(n^2 * 2^n)$, and Branch and Bound also $O(n!)$. In contrast, Genetic Algorithms are classified as heuristic methods, and their complexity depends on the specific parameters chosen.

## EXPERIMENTAL SETTINGS

In the domain of evolutionary computation, the research outlined by *Wei et al. (2019)* underscores the crucial significance of appropriately configuring algorithmic parameters to tackle NP-hard optimization problems. Concentrating specifically on the Traveling Salesman Problem (TSP), the investigation underscores the significance of carefully selecting suitable parameter values and thoroughly assessing the impact of diverse TSP instances. A fixed population size of 100 and 20,000 generations is maintained, coupled with a crossover rate of 0.08 and a mutation rate of 0.01, as detailed in the study. Additionally, *Triana, Bucheli & Garcia (2020)* aligns its parameter choices with previous research, adhering to a consistent population size of 100 and a mutation rate of 0.01. The rationale behind these parameter selections is deeply rooted in an extensive exploration of evolutionary computation approaches for TSP problem-solving, as elucidated by *Wei et al. (2019)*. Consequently, in this current study, the parameter setup mirrors a similar strategy while also evaluating different city sizes to further broaden the analysis.

In the experimental setup, we established several parameters to ensure rigorous testing of our proposed approach (see Table 1). To determine the appropriate population size and mutation rate, we drew insights from prior research in the field of evolutionary computation focused on addressing the TSP, these parameters were set as follows: a population size of 100, a mutation rate of 0.1, and a stop criterion of 100 iterations (*Wei et al., 2019*). These values were aligned with the configuration of the traditional genetic

**Table 1 The parameters of experimental settings.**

| Parameter | Value |
| --- | --- |
| Numbers of cities | [5, 25, 100, 500, 1,000] |
| Population size | 100 |
| Mutation rate | 0.1 |
| Stop criterion | 100 |

algorithm, serving as our baseline. In addition, the Watts–Strogatz evolutive model was tested (*Watts & Strogatz, 1998*).

To comprehensively assess the performance of our proposed approach, we selected a diverse set of benchmark instances from the TSP, which covered a wide range of cities, varying from 5 to 1,000. These instances covered a range of city quantities. 5, 25, 100, 500 and 1,000. This broad selection allowed us to evaluate the scalability and effectiveness of our method across different problem sizes. Our experimental design emphasized robustness by repeating experiments multiple times with distinct random seeds, accounting for the inherent stochasticity of evolutionary algorithms. The reported results represent the average outcomes of more than 100 independent runs, focusing on key metrics such as average fitness value and convergence speed.

The experiments were executed on a dedicated computer system with specific hardware and software configurations. The hardware setup comprised a 2-core Xeon 2.2 GHz processor, 13 GB of memory, and a 33 GB hard disk. For software tools, we employed Python 3.7, implemented the TSP algorithm, and utilized Spyder 3.3.3 and the Networkx 2.4 library to model complex networks. The entire implementation and testing process, including the source code, can be accessed in our repository hosted at GitHub. The implementation and testing of the model can be found in the following repository: github.com/jodatm/complex_networks_in_EA.

## RESULTS

The results represent the averages of 100 experiments conducted and each experiment ran 100 iterations. Figure 4 illustrates the performance comparison of traditional evolutionary algorithms (blue), Watts–Strogatz evolutive model (yellow), and Erdős–Rény evolutive model (red). In Fig. 2A experiments were conducted with 25 cities, and the results demonstrate that the Erdős–Rény model outperforms the other models. Figures 2B–2D represent experiments conducted with 100, 500, and 1,000 cities, respectively, showing consistent performance where the proposed model yields superior results. These findings indicate that the proposed model exhibits better performance than the traditional evolutionary algorithm in the fitness value.

The experimental tests conducted on the proposed algorithms indicate that incorporating neighbor selection based on better fitness values could be the optimal alternative. As suggested by *Triana, Bucheli & Solarte (2022)*, the best fitness was obtained using the best neighbor model, which involves selecting the neighbor through the roulette

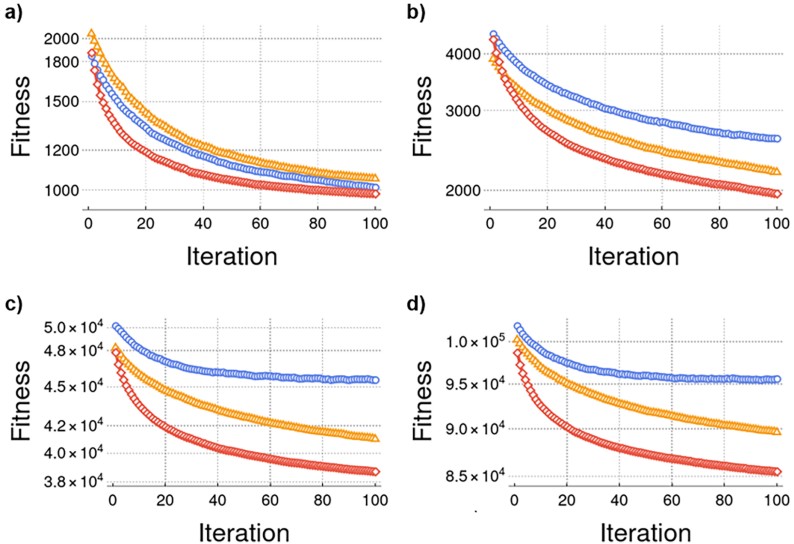

**Figure 4** (A–D) Average performance in TSP problem.

method. The intention is to favor the selection of the neighbor with better performance (fitness). In Fig. 4A, which presents the same number of cities as in the study conducted by *Triana, Bucheli & Garcia (2020)*. The result demonstrates that the Erdős–Rény evolutionary model converges earlier and accelerates faster than the other models. Convergence of the proposed Watts–Strogatz evolutionary model and traditional algorithms is expected to occur after a certain number of iterations.

As shown in Fig. 4, the proposed approach converges to minimize the Fitness value as the iterations progress. This convergence shows an improvement in the algorithm's efficiency for facing the TSP problem. Moreover, when we evaluated the performance of 25 cities, the convergence rate was similar for all the tested approaches. However, as the number of cities grows, the approach proposed in this article outperforms the traditional evolutionary algorithms and Watts–Strogatz approaches (see Figs. 4C and 4D). The results indicate that our proposal converges in all experiments from 25 to 1,000 cities and is feasible for dealing with the TSP problem.

We analyzed the impact of the computational cost of the proposed model compared to the original strategy. Figure 5 showcases the traditional evolutionary strategy applied to the TSP problem with a range of 5 to 1,000 cities, where each point represents a specific instance. The figure illustrates the traditional evolutionary algorithms (blue), the Watts–Strogatz evolutionary model (yellow), and the Erdős–Rényi evolutionary model (red). It is evident that the computational cost of the proposed models, measured in seconds of execution time, follows a logarithmic scale as the instance size increases. Moreover, we examined how the behavior changes as the order of magnitude increases and observed it over iterations. The model consistently demonstrates an advantage in terms of fitness and exhibits a logistic relationship with CPU times. The results demonstrated that the performance of the proposed model, particularly employing the Erdős–Rényi network

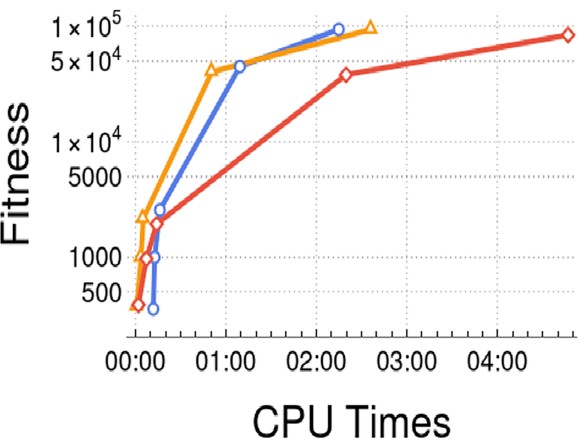

**Figure 5 Performance in TSP problem *vs* CPU times.**

approach, outperformed the traditional algorithm in achieving satisfactory fitness within a reasonable timeframe.

In regard to the Erdős–Rény evolutionary model, the computational complexity is not directly derivable from the provided data. However, by observing the increase in execution time as the input size (number of cities) grows, it is evident that the algorithm exhibits a significant increase in runtime with larger instances. This suggests that the complexity of the proposal likely grows with the input size and that it may require further analysis to determine its theoretical computational complexity, potentially involving regression or more extensive testing with various input sizes.

The effectiveness of a genetic algorithm in tackling the TSP is intricately linked to the careful selection of its parameters. Among these parameters, the population size, mutation rate, crossover rate, and the number of generations play pivotal roles in shaping the algorithm's behavior and overall performance. To illustrate, a larger population size fosters exploration but concurrently results in increased computational time, while a higher mutation rate assists in evading local minima but may potentially lead to divergence if set excessively. Understanding and optimizing these parameters represent crucial steps towards enhancing the efficiency and efficacy of the genetic algorithm for solving the TSP. Regarding the aspects described below and the theoretical computational complexity O, further analysis and research is necessary to precisely determine its evaluation, leaving it as a potential avenue for future investigation and study.

## DISCUSSION

The TSP has been commonly addressed using heuristic algorithms. Previous research has applied several complex network properties to the TSP, such as Community Detection (*Liu, Liu & Jiang, 2014*) and PageRank Distance (*Jiang, Liu & Wang, 2016*; *Triana, Bucheli & Garcia, 2020*). In this article, we proposed an evolutionary algorithm guided by Erdős–Rényi complex networks to regulate population crossovers, enhancing candidate solution refinement across generations. In all the conducted experiments, selecting the neighbors with the highest fitness values consistently resulted in the best outcomes. This selection

approach promotes competition among individuals in the population as they strive to improve their fitness, which likely explains the observed behavior. Experimental results show that the Erdős–Rény evolutionary model is a feasible strategy to address the TSP problem.

The main contribution of this proposal is the enhancement of the crossover and selection process in the genetic algorithm by incorporating the Erdős–Rény complex network. Thus, parents are selected based on the characteristics of the neighborhood surrounding that individual in the network. Maintaining better solutions and increasing the crosses between better solutions by the parent-child relationships and the groups of solutions or clusters. To our knowledge, this is the first proposal that uses Erdős–Rény networks for addressing the TSP problem.

Integrating an Erdős–Rény network allows us to investigate the computational aspects of constructing the network model. Compared to a traditional algorithm and Watts–Strogatz network model, the Erdős–Rényi random graph has traditionally represented complex network topologies. Notably, the Erdős–Rényi random graph possesses the distinctive characteristic that many of its intriguing properties can be analytically expressed. This distinguishes it from other complex network graph models, which often present challenges or limitations in computational analysis. Furthermore, it opens new research frontiers, the analytical study of evolutionary network models (*Jamakovic & Van Mieghem, 2008*).

The Watts-Strogatz network model (*Watts & Strogatz, 1998*), which involves rewiring edges, is extensively utilized in the field of network research. It is commonly used to evaluate the impact of the small-world effect on dynamic processes occurring within the network. The Watts-Strogatz model effectively captures both regular and random features observed in real-world networks. However, the Erdős-Rényi model exhibits notable distinctions in degree distribution, path length, clustering coefficient, and the presence of giant components. The characteristics of the topology of the Erdős-Rényi graph are transferred to the grouping of solutions. Then, while the number of iterations grew, the number of communities of better solutions grew, too. Therefore, the time expected to solve the TSP problem is reduced by crossing the neighborhood of better solutions between them. Finally, the mutation process generates diversity in the communities, which is used to obtain better solutions.

The experimental results showed that the Erdős-Rényi models performed better than a traditional evolutionary algorithm and Watts-Strogatz network model. The experiments highlight the consistency of the results obtained by the proposed model. In addition, this model scales logarithmically the CPU times over the growth of cities, where the size of a set of cities is evaluated from five, 25, 100, 500, and 1,000 cities. Comparing the same instances of the optimization problem to other studies (*Triana, Redondo & Bucheli, 2019*; *Jiang, Liu & Wang, 2016*), the proposed algorithm obtains better performance than those studies. Furthermore, the fitness value and the time spent (CPU times) is minimized in a better way by the Erdős-Rényi model.

Future research will thoroughly examine each Erdős-Rényi model's characteristics to understand better how they contribute to enhanced performance. Therefore, complex

networks can effectively regulate the decision-making process, similar to optimization problems. This study highlights the critical role of the network structure, the Erdős-Rényi model *vs* the Watts-Strogatz model, in adding value to the decision-making process.

## CONCLUSIONS

The Erdős-Rényi evolutive model consistently outperforms the baseline model and Watts-Strogatz network mode, which aligns with previous research findings, where the complex network to guide an evolutionary computation algorithm proves to be reasonable based on the obtained better results.

The relation between time and fitness performance shows the scalability of the TSP. The results are promising and highlight the model's impact on larger-size problems. Moreover, the potential additional cost of incorporating complex network dynamics into the evolutionary algorithm is insignificant, as the results indicate. This fact is mainly attributed to the computational advantage of local computations attributable to the selection process, father-son relationships, and communities of solutions. The computational advantages are due to the Erdős-Rényi networks and their topology characteristics: Poissonian degree distribution, short path length, small clustering coefficient, and the presence of giant components.

### Funding
The authors received no funding for this work.

### Competing Interests
The authors declare that they have no competing interests.

### Author Contributions
- Víctor A. Bucheli conceived and designed the experiments, performed the experiments, analyzed the data, performed the computation work, prepared figures and/or tables, authored or reviewed drafts of the article, and approved the final draft.
- Oswaldo Solarte Pabón conceived and designed the experiments, performed the computation work, prepared figures and/or tables, authored or reviewed drafts of the article, and approved the final draft.
- Hugo Ordoñez conceived and designed the experiments, performed the experiments, analyzed the data, prepared figures and/or tables, and approved the final draft.

### Data Availability
The implementation of evolutionary algorithm that integrates Erdős–Rény Complex Networks to regulate population crossovers is available at GitHub and Zenodo:

- https://github.com/jodatm/erdos-reny.

- OSWALDO SOLARTE. (2023). Evolutionary Algorithms Guided by Erdős–Rényi Complex Networks. https://doi.org/10.5281/zenodo.8414400.

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
