# Peer review of "Evolutionary algorithms guided by Erdős–Rényi complex networks"

_PeerJ Computer Science, doi:10.7717/peerj-cs.1773_

## Round 0.1 · original submission · Major Revisions

Please address the comments of the two reviewers. They both recommend extending the experimental evaluation of the proposed method and explaining why the new algorithm performs better in the context examined.

Reviewer 1 ·

Basic reporting

This work aims to design an evolutionary algorithm that integrates ErdQsñRÈny Complex Networks to regulate population crossovers. The population of the evolutionary computation is represented as a complex network, where nodes represent individual solutions and links represent crossovers among solutions. The algorithm enhances solutions by creating new links among nodes, guiding population dynamics, and optimizing problem-solving processes. The results indicate that the strategy guided by a complex dynamic network outperforms the traditional algorithm. However, in my opinion, there are major issues that need to be addressed to allow a publication in Journal:

1. The authors integrate ErdQsñRÈny Complex Networks to regulate population crossovers and implement it. The authors see that the integration obtains better results regarding quality and running time in experiments. However, the authors do not analyze why this integration helps the algorithm to be better. The results need to be considered deeper.
2. The authors compare the proposed algorithm with the traditional one (very basic) to demonstrate the efficiency of the proposed algorithm. It is not realistic. It needs to be compared with many state-of-the-art schemes in the literature.
3. The experimental results are simple. It is not enough to show the efficiency of the proposed algorithm. The authors should analyze more detail and explain why the proposed algorithm obtain better solutions. Authors should see inside the proposed algorithm to demonstrate its efficiency

Experimental design

The experimental results are simple. It is not enough to show the efficiency of the proposed algorithm.

Validity of the findings

1. The authors integrate ErdQsñRÈny Complex Networks to regulate population crossovers and implement it. The authors see that the integration obtains better results regarding quality and running time in experiments. However, the authors do not analyze why this integration helps the algorithm to be better. The results need to be considered deeper.
2. The authors compare the proposed algorithm with the traditional one (very basic) to demonstrate the efficiency of the proposed algorithm. It is not realistic. It needs to be compared with many state-of-the-art schemes in the literature.
3. The experimental results are simple. It is not enough to show the efficiency of the proposed algorithm. The authors should analyze more detail and explain why the proposed algorithm obtain better solutions. Authors should see inside the proposed algorithm to demonstrate its efficiency

Cite this review as

Reviewer 2 ·

Basic reporting

This work aims to design an evolutionary algorithm that integrates ErdQs3Rény Complex Networks to regulate population crossovers. The topic of this work is interesting, I have the following suggestion to further improve the paper.
1. The meaning of “complex networks” used in combinatorial optimization problem is unclear. It is mentioned that “The vertices (nodes) in the graph correspond to individuals or the population of solutions, while the edges (links) represent the connections between individuals.” Does “complex networks” refer to this relationship in all combinatorial optimization problems? If so, please describe in more detail; If not, please introduce what complex networks stands for in different studies.

2. The description of the TSP problem is too long and the description of the method is too short. Please explain the basic principles or processes of the Erdos-Reny network and the Watts-Strogatz network model for comparison. Please also describe the process of the traditional genetic algorithm used for comparison, and the corresponding sources.

3. What is the definition of “Fitness value” for this problem and what is the meaning of “k” in Algorithm 1?

4. The experimental results in this paper demonstrate the effect of different strategies on fitness value, are there other comparisons to evaluate the effect of different models?

5. Please analyze the reasons for the results of this experiment. (The effect of differences in the different models on solving the problem, or the contribution of the principles of the different models to solving the problem)

6. There is no content for the Acknowledgement in this paper, please continue to complete or delete the section.

Experimental design

no comment

Validity of the findings

no comment

Additional comments

no comment

Cite this review as

---

## Round 0.2 · Major Revisions

I have now received the reports of the two reviewers. As you will see below, one of them still has a few comments for you to consider.

Reviewer 1 ·

Basic reporting

The authors tried to address some comments of the reviewer. The revised manuscript is quite better than the original one. However, I still have some comments:
- How to choose parameters. You said that the values were aligned with the configuration of the traditional genetic algorithm, serving as our baseline. You should cite the paper with the same settings.
- You should analyze the theoretical complexity of the algorithm as well as the running time with the other algorithms?.
- You should show the convergence trends of the proposed algorithm to show the efficiency of the algorithm.

Experimental design

- You should show the convergence trends of the proposed algorithm to show the efficiency of the algorithm.

Validity of the findings

no comment

Cite this review as

Reviewer 2 ·

Basic reporting

The authors addressed all my concerns I raised; the paper can be accepted for publication.

Experimental design

no comment

Validity of the findings

no comment

Additional comments

no comment

Cite this review as

---

## Round 0.3 · accepted · Accept

The reviewer has recommended that I accept the manuscript for publication as is, and I am happy to do so. Thank you for your contribution to PeerJ Computer Science!

Reviewer 1 ·

Basic reporting

The authors have revised all comments from reviewers.

Experimental design

OK

Validity of the findings

OK

Additional comments

OK

Cite this review as